# In Vitro Modeling of Reoxygenation Effects on mRNA and Protein Levels in Hypoxic Tumor Cells upon Entry into the Bloodstream

**DOI:** 10.3390/cells9051316

**Published:** 2020-05-25

**Authors:** Kai Bartkowiak, Claudia Koch, Sebastian Gärtner, Antje Andreas, Tobias M Gorges, Klaus Pantel

**Affiliations:** Department of Tumor Biology, University Medical Center Hamburg-Eppendorf, Martinistr. 52, 20246 Hamburg, Germany; k.bartkowiak@uke.de (K.B.); c.koch@uke.de (C.K.); sebastian_gaertner@mail.de (S.G.); a.andreas@uke.de (A.A.); t.gorges@uke.de (T.M.G.)

**Keywords:** dissemination, breast cancer, hypoxia, reoxygenation

## Abstract

Background: Solid epithelial tumors like breast cancer are the most frequent malignancy in women. Circulating tumor cells (CTCs) are frequently released from hypoxic areas into the blood, where CTCs face elevated oxygen concentrations. This reoxygenation might challenge the use of CTCs for liquid biopsy. Methods: We modeled this situation in vitro using the breast cancer cell lines—MCF-7, MDA-MB-468, MDA-MB-231—and the cell line BC-M1 established from DTCs in the bone marrow. Cells were cultured under hypoxia, followed by a reoxygenation pulse for 4 h, reflecting the circulation time of CTCs. Analyzed were gene products like EGFR, ErbB-2, EpCAM, PD-L1 on mRNA and protein level. Results: mRNAs of *erbb2* or *pdl1* and protein levels of PD-L1 displayed significant changes, whereas ErbB-2 protein levels remained constant. The strongest discrepancy between protein and mRNA levels under hypoxia was observed for EGFR, supporting the idea of cap-independent translation of *egfr* mRNA. Analyses of the phosphorylation of AKT, Erk 1/2, and Stat3 revealed strong alterations after reoxygenation. Conclusions: CTCs reaching secondary sites faster than reoxygenation could alter the mRNA and protein levels in the cells. CTC and DTC with high PD-L1 levels might become quiescent under hypoxia but were easily reactivated by reoxygenation.

## 1. Introduction

Molecular analysis of circulating tumor cells (CTCs) allows novel insights into the biology of cancer metastasis and provides diagnostic information as liquid biopsy [1]. In breast cancer, the detection of CTCs in the blood and disseminated tumor cells (DTC) at secondary sites (e.g., the bone marrow) are useful predictors of distant metastasis in cancer patients [2,3,4].

For the detection of CTC and DTC, epithelial markers of the keratin family-like cytokeratin 8 and 19 are established [2]. In addition, cytokeratin 5 can be applied for the identification of triple-negative breast cancers, which contains the group of epithelial EGFR (epithelial growth factor receptor) overexpressing basal-like breast cancers [5]. On the other hand, tumor cells with mesenchymal attributes that have passed an epithelial-mesenchymal transition (EMT) express proteins like vimentin [6].

One factor that can promote the acquisition of mesenchymal attributes and dissemination of tumor cells is hypoxia [7,8]. In primary tumors, hypoxia occurs when the tumor cells grow faster than novel blood vessels are formed, leading to oxygen deficiencies in the microenvironment. Generally, tumor cells that are adjacent to blood vessels receive oxygen and other nutrients like amino acids. However, in tumors, blood vessels may strongly vary in diameter, show arteriolar-venous shunts, the wall of the blood vessels is compromised, or the spatial and temporal blood flow is sporadic. This may lead to tumor hypoxia, even adjacent to capillaries [9]. Unlike other tissues, hypoxia is a physiological condition in the hematopoietic stem cell niche in the bone marrow, where values of only 1% O_2_ (hypoxia) are detected [10,11]. DTC in the bone marrow, as a frequent metastatic site in breast, prostate, and lung cancer, appears to prefer the hematopoietic stem cell niches, which are located in the most hypoxic areas [12]. Recently, we have shown that DTC in bone marrow has adapted to these microenvironment conditions by the activation of the unfolded protein response (UPR) proteins like the 78 kDa glucose-regulated protein (Grp78) [13,14].

Tumor cells activate three major adaptation programs to cope with hypoxic stress. Metabolic adaptation is governed by the hypoxia-inducible factor 1-alpha (HIF-1α), the adaptation of the global protein synthesis is controlled by the mammalian target of rapamycin (mTOR), and the UPR maintains proper folding of nascent proteins in the endoplasmic reticulum [15,16]. Moreover, hypoxia can result in an abrogation of the global protein translation, leading to the accumulation of mRNAs [15,17]. However, some mRNAs can be translated via internal ribosome entry sites (IRES) by cap-independent protein translation, even when the global protein synthesis is shut down. For several tumor-relevant gene products like Grp78, estrogen receptor alpha (ER-α), HIF-1α, or vimentin, IRES has been detected [18], suggesting that these proteins are synthesized under hypoxia.

Obviously, it is almost impossible to determine how long CTCs in cancer patients have been subjected to hypoxic conditions in the primary tumor or metastatic sites before their release into the bloodstream. The previously reported detection of vascular endothelial growth factor/HIF-1α double-positive CTC [19] in the blood of breast cancer patients suggests that these cells have responded to prior hypoxic conditions in the tumor tissue and may experience a pulse of reoxygenation after release into the blood circuit [20], as simulated in our present in vitro model. Due to a potential uncoupling of the mRNA from protein levels, it is not clear if mRNA levels of a specific gene product indeed correlate with its protein level. Moreover, it is considered that detected mRNA and protein (phosphorylation) levels in CTC rather reflect a response to reoxygenation in the blood circuit than the expression profile of the tumor cells at-site, which would limit the use of CTC for liquid biopsy.

Here, we developed an in vitro model to determine the changes in the cellular levels of breast cancer-relevant gene products at the mRNA and (phospho)protein in tumor cells exposed to hypoxia and subsequent reoxygenation. This model represented a first experimental approximation to identify gene products that might be altered when tumor cells leave hypoxic tissues and enter the well-oxygenated blood circuit as CTC.

## 2. Materials and Methods

### 2.1. Antibodies used for Western Blot

Antibodies were purchased from the following suppliers: 

Abcam, Cambridge, United Kingdom: Anti-ErbB-2 (CB11) antibody, mouse monoclonal, dilution 1:500. Anti-CXCR4 antibody, rabbit polyclonal, dilution 1:4000. Anti-Cytokeratin 5 antibody [XM26], mouse monoclonal, dilution 1:5000. Anti-EpCAM antibody [VU-1D9], mouse monoclonal, dilution 1:1000. Anti-estrogen receptor alpha [SP1], rabbit monoclonal, dilution 1:1000. 

BD Pharmingen, Erembodegem, Belgium: Anti-Vimentin antibody, mouse monoclonal clone RV202, dilution 1:5000 (MDA-MB-468, MCF-7), other cell lines 1:10^7^.

Epitomics, Burlingame, CA, USA: anti-E-Cadherin antibody, rabbit monoclonal clone EP700Y, dilution 1:20,000. 

Millipore, Schwalbach, Germany: Anti-Cytokeratin 19, clone BA 17, mouse monoclonal, dilution 1:100,000.

Novus Biologicals, Littleton, CO, USA: Anti-N-Cadherin antibody (EPR1792Y), rabbit monoclonal, dilution 1:100,000.

Progen, Heidelberg, Germany: anti-Keratin K8 clone K8.2, mouse monoclonal, dilution 1:10,000.

Santa Cruz Biotechnology, Santa Cruz, CA, USA: Anti-EGFR (1005) antibody, rabbit polyclonal, dilution 1:250 or 1:5000. 

Cell Signaling Technology, Danvers, MA, USA: anti-AKT antibody, rabbit polyclonal, dilution 1:2000. Anti-phospho-Akt (Ser473) (193H12) antibody, rabbit monoclonal, dilution 1:1000. Anti-p44/42 MAPK (Erk1/2) antibody, rabbit polyclonal, dilution 1:2000. Anti-phospho-p44/42 MAPK (Erk1/2) (Thr202/Tyr204) antibody, rabbit polyclonal, dilution 1:3000. Anti-Stat3 antibody, rabbit polyclonal, dilution 1:1000. Anti-phospho-Stat3 (Tyr705) antibody, rabbit polyclonal, dilution 1:1000. Anti-alpha-Tubulin (11H10) antibody, rabbit monoclonal, dilution 1:10,000. Anti-BiP (C50B12) antibody, rabbit monoclonal (BiP is a synonym for Grp78), dilution 1:1000. Anti-PD-L1 (E1L3N) XP antibody, rabbit monoclonal, dilution 1:2000. Anti-CD44 antibody, rabbit polyclonal, dilution 1:5000. Anti-HSPA8 (D12F2), rabbit monoclonal, dilution 1:2000. Anti-p70 S6 kinase, rabbit polyclonal, dilution 1:10,000. Anti-phospho-p70 S6 kinase (T389), mouse monoclonal, dilution 1:1000. Anti-HIF-1α (D2U3T), rabbit monoclonal, dilution 1:10,000.

Quantitative RT-PCR. The mRNA levels of *hspa5* (Grp78), *cd274* (PD-L1), *vim* (vimentin), *egfr* (EGFR), *epcam* (EpCAM), *erbb2* (ErbB-2), and esr1 (ER-α) were quantified. The values were normalized to the values of the housekeeping gene *hspa8* (Hsc70). 

RNA was isolated using the NucleoSpin RNA II kit (Machery-Nagel, Düren, Germany), followed by cDNA synthesis (First Strand cDNA Synthesis Kit, Thermo Fisher, Waltham, MA, USA) according to manufacturer’s instructions. Primers against *Vimentin* (fw_GAGAACTTTGCCGTTGAAGC, rev_TCCAGCAGCTTCCTGTAGGT), *EGFR* (fw_CAGCGCTACCTTGTCATTCA, rev_TGCACTCAGAGAGCTCAGGA), *EpCAM* (fw_GCTGGTGTGTGAACACTGCT, rev_ACGCGTTGTGATCTCCTTCT), *ErbB2* (fw_TGCCTGTCCCTACAACTACC, rev_CAGACCATAGCACACTCGG), and *HSC70* (fw_GAGCAAGGAAGACATTGAACG, rev_ATGACACCTTGTCCCTCTGC) were designed using the Primer3 software [21]. Primers targeting mRNA of *GRP78* (fw_CGACCTGGGGACCACCTACT, rev_TTGGAGGTGAGCTGGTTCTT) [22] and *ERα* (fw_GCATTCTACAGGCCAAATTCA, rev_TCCTTGGCAGATTCCATAGC) [23] were extracted from literature. *PDL1* primers (fw_AAGAAAAGGGAGAATGATGGATGTG, rev_GCTGGATTACGTCTCCTCCAA) were kindly provided by Sonja Mader (Institute for Tumor Biology). The qPCR was performed in a CFX96 Touch Real-Time PCR Detection System (Bio-Rad, Hercules, CA, USA) using Maxima SYBR-Green fluorescent dye (Thermo Fisher Scientific, Waltham, MA, USA). Amplification was performed under the following conditions: after an initial denaturation step (10 min at 95 °C), 40 amplification cycles were carried out, consisting of denaturation at 95 °C for 30 s, annealing at 60 °C for 30 s, and elongation for 30 s at 72 °C. A final elongation step at 72 °C (10 min) was followed by a melting curve analysis and storage of the samples at 4 °C. Data analysis was performed using the CFX Manager Software (BioRad, Feldkirchen, Germany). Relative gene expression was calculated from data sets according to the comparative C_T_ (ΔΔC_T_) method [24]. In brief, the first amplification cycle displaying a significant increase of fluorescence signal over background level was defined as threshold cycle; C_T_ data were normalized by subtracting the C_T_ value of *hspa8* from the C_T_ of the target gene, resulting in a Δ C_T_ value. The ΔΔC_T_ was then calculated as follows: ΔΔC_T_ = Δ C_T Treatment_ − Δ C_T Control_. Finally, the ΔΔC_T_ was converted to fold change using the formula 2^−ΔΔC^_T_.

### 2.2. Cell Lines and Culture Conditions

Cell lines were cultured at 37 °C in a humidified environment. Cell lines cultured in DMEM were kept in the presence of 10% CO_2_, and the cell lines cultured in RPMI were kept in the presence of 5% CO_2_. The remaining gas mixture was atmospheric air. MCF-7 (from ATCC, 2005), MDA-MB-231, and MDA-MB-468 (both from Cell Lines Service, Eppelheim, Germany, 2007) were cultivated in DMEM with 10% FCS and 2 mM L-glutamine. Authentication (last test): MCF-7/MDA-MB-231 (02/2014); MDA-MB-468 (05/2015). Authentication was done by Multiplexion, Heidelberg, Germany by SNP-Profiling. BC-M1 is a DTC cell line from the bone marrow of a breast cancer patient and was generated in 1994 and authenticated by Klaus Pantel [25,26]. The last authentication was done on May 2015 by immunofluorescent double staining for pancytokeratin/vimentin. BC-M1 was cultured with 10% of oxygen. These conditions referred as to “standard cell culture condition” in this work. Cultivation of the cell lines under 1% or 10% O_2_ (hypoxia) was performed using the incubator Heracell 15 (Thermo Fisher Scientific, Waltham, MA, USA). The oxygen partial pressure was adjusted by N_2_.

### 2.3. Densitometric Analysis

Western blot analyses were performed, as described in [14]. For the analysis of p70 S6 kinase, phospho-p70 S6 kinase (T389), and HIF-1α, 8% separation gels were used. The applied antibodies are specified in supporting information. RNA and protein were collected from different cell culture flasks in parallel biological triplicates.

### 2.4. Quantitative RT-PCR

For quantitative mRNA analysis, the levels of the housekeeping gene *hspa8* (Hsc70) were used for normalization. RNA was isolated using the NucleoSpin RNA II kit (Machery-Nagel, Düren, Germany), followed by cDNA synthesis (First Strand cDNA Synthesis Kit, Thermo Fisher, Waltham, MA, USA). The primers for PD-L1 were kindly provided by Sonja Mader. The qPCR was performed in a CFX96 Touch Real-Time PCR Detection System (Bio-Rad, Hercules, CA, USA). The relative gene expression was calculated from datasets according to the comparative C_T_ (ΔΔC_T_) method [24].

## 3. Results

### 3.1. The Response of Hypoxia Response Proteins to Hypoxia and Reoxygenation

To investigate the effect of reoxygenation on tumor cells, we subjected breast cancer cells from different cell lines to hypoxia (1% O_2_) for 14 days, reflecting hypoxic conditions prior to entering the blood circuit (starting condition of CTCs). Subsequently, cells were subjected to a reoxygenation pulse of 10% O_2_ for 4 h, which mimicked the situation of CTCs in the bloodstream. The half-life of CTCs in the blood of cancer patients is estimated to be 2–3 h, and after 4 h, two-thirds of the CTC departed from the blood circuit [27].

The selected oxygen concentration and cultivation time are suitable model conditions for hypoxia in breast cancer [14,28]. For reoxygenation, we selected 10% O_2_ as an average value for larger human blood vessels [20], and the time frame was adapted to the life span of CTC in the blood [27]. Analyzed were representative breast cancer cell lines for the estrogen receptor-positive (MCF-7), basal-like (MDA-MB-468), and triple-negative (MDA-MB-231) phenotype. In addition, we used BC-M1 as a well-characterized breast cancer DTC cell line from the bone marrow [14,26]. We investigated the response of three key proteins of the major hypoxia response programs and analyzed cells cultured under standard cell culture conditions (S), hypoxia (H/14d), and reoxygenation (R/14d 4h) (Figure 1).

Grp78 of the UPR is involved in the folding of nascent proteins in the endoplasmic reticulum and can mediate resistance against chemotherapy [29]. Compared with the standard culture conditions, Grp78 protein levels were decreased under hypoxia and reoxygenation in MCF-7, whereas for reoxygenation in MDA-MB-468, a slight induction of Grp78 levels was detected in comparison with the hypoxic conditions. In MDA-MB-231 and BC-M1, only minor changes in the Grp78 levels upon hypoxia and reoxygenation were detected.

Next, the master regulator of metabolic adaptation to hypoxia HIF-1α was investigated. HIF-1α is regulated on the protein level by Hippel–Lindau tumor suppressor (pVHL) E3 ligase complex [30]. Here, we observed for all cell lines downregulation of HIF-1α after 14 days of hypoxia compared with the standard culture conditions. Recovery of the HIF-1α levels after reoxygenation was detected for MDA-MB-468 and MDA-MB-231 and to a lesser extent for BC-M1, but not for MCF-7.

On the protein phosphorylation level, we investigated the p70 S6 kinase on T389. Under normal conditions, mTOR stimulates protein synthesis by phosphorylation of the p70 S6 kinase, leading to the promotion of protein translation [15]. For less transformed breast cancer cells, it has been reported that the phosphorylation of T389 is reduced under hypoxia, whereas highly transformed cells maintain the phosphorylation of T389 [31]. Compared with normal conditions, we observed decreased phosphorylation of T389 under hypoxia and reoxygenation in MCF-7 and BC-M1, while the signals in MDA-MB-231 were too low for confident analysis. In contrast, a strong induction of phosphorylation on T389 was observed in MDA-MB-468 upon reoxygenation (Figure 1).

### 3.2. Cellular Responses to Hypoxia and Reoxygenation in Breast Cancer-Relevant Proteins

The epidermal growth factor receptor EGFR and its heterodimerization partner ErbB-2 are receptor tyrosine kinases and prominent drivers of malignant transformation as well as therapeutic targets [32]. Similarly, estrogen receptor alpha (ER-α)-positive breast cancers are treated with hormone therapy against ER-α [33]. The immune checkpoint regulator programmed death-ligand 1 (PD-L1) is regarded as a novel promising therapeutic target and has been detected on CTC [34], and CXCR4 is an important receptor for tumor cell dissemination to the bone marrow.

For EGFR, we detected elevated levels in all analyzed cell lines upon hypoxia, with the exception of MDA-MB-231 (Figure 2 and Figure 3). In the case of MDA-MB-231, we detected a significant change in the EGFR levels upon reoxygenation (Table 1). For ErbB-2, we observed an induction under hypoxia in MDA-MB-468, which was maintained under reoxygenation, whereas for the other investigated cell lines, only minor changes were observed. The only ER-α positive cell line we analyzed was MCF-7. Compared with the standard cultivation, we observed a decrease of the ER-α levels to 23% under hypoxia and a slight increase to 33% under reoxygenation conditions. For the ER-α negative cell lines, no induction of this protein was observed.

PD-L1 was induced in MDA-MB-231 under hypoxia, followed by a significant (*p* = 0.004) decreased levels under reoxygenation (Figure 2, Table 1) compared with hypoxia. Notably, the DTC cell line from the bone marrow of a breast cancer patient BC-M1 exhibited, even under standard conditions, high levels of PD-L1 (Figure 2). Under hypoxia, these levels increased 3.6-fold compared with standard culture conditions (Figure 3). Upon reoxygenation, the PD-L1 levels significantly (*p* = 0.024) decreased in comparison with the hypoxic conditions (Figure 3, Table 1).

We observed only minor changes in the protein levels of CXCR4 under hypoxia compared with the standard conditions in the cell lines, and changes in response to reoxygenation were not observed for all investigated cell lines (Figure 2).

### 3.3. Changes of Epithelial Differentiation Marker Proteins under Hypoxia and Reoxygenation

For the detection of CTC and DTC, epithelial differentiation marker proteins of the keratin family or the transmembrane protein EpCAM are applied [2]. Since it is assumed that CTC and DTC may reduce epithelial and acquire mesenchymal attributes, it is possible that cells with a strong manifestation of mesenchymal attributes become undetectable by epithelial marker proteins. Such cells are considered to be the actual metastasis founder cells [35]. For the characterization of the epithelial/mesenchymal attributes, additional proteins like E-cadherin, N-cadherin, CD44, or vimentin are applied [1].

For the proteins of the keratin family, only minor changes in the expression level under hypoxia and reoxygenation were observed (Figure 2). Indeed, a certain induction of CK5 in MDA-MB-468 was observed. On the other hand, the very high EpCAM levels were reduced to 43% and 34% of the standard culture value in MDA-MB-468 under hypoxia and reoxygenation, respectively (Figure 3). Moreover, on the induction of the marker proteins for mesenchymal cells, vimentin was observed in MDA-MB-468, whereas for the strongly mesenchymal cell lines, no changes in the vimentin levels were observed (Figure 2).

No changes in the protein levels of N-cadherin and E-cadherin, as well as for CD44, under hypoxia and reoxygenation in the investigated cell lines were observed.

### 3.4. Cellular Responses to Hypoxia and Reoxygenation on mRNA Level

We selected from our analysis on protein level a set of gene products that responded to hypoxia for a closer analysis on the mRNA level (Figure 3). Grp78 was selected as a protein of the hypoxia response program UPR for whose *grp78* mRNA IRES was detected. The changes in the levels of the *grp78* mRNA under hypoxia and reoxygenation were heterogeneously distributed in the cell lines. While for MCF-7, no changes in the mRNA levels were observed, in MDA-MB-468, the levels of the *grp78* mRNA under hypoxia increased 2.9-fold and fell down 0.85 under reoxygenation (Figure 3). The reduction of the *grp78* mRNA levels under reoxygenation (compared with hypoxia) was also observed in BC-M1, though without prior induction under hypoxia. In contrast, the *grp78* mRNA was reduced to 0.43 compared with the standard culture conditions in MDA-MB-231 but reached a value of 1.14 upon reoxygenation (Figure 3).

Another gene product containing IRES is *er-α.* We detected *er-α* mRNA in all investigated cell lines, and we also detected changes in the *er-α* mRNA. In MDA-MB-468 and BC-M1, the *er-α* mRNA was downregulated after reoxygenation compared with the standard culture conditions, whereas in MDA-MB-231, the *er-α* mRNA reached an induction of 4.0 compared with the standard culture conditions. In contrast, the third gene product containing IRES–*vim (vimentin)*–showed no detectable changes on the mRNA level in all investigated cell lines (Figure 3). Another gene product that was detectable on the mRNA level, but not on the protein level, was *epcam.* Similarly, only minor changes for the *epcam* and for the *pdl1* mRNA were detected under hypoxia and reoxygenation in the investigated cell lines.

Remarkable differences in the response of the *egfr* and the *erbb-2* mRNA were observed even though both gene products belong to the same family. While only minor changes in the *egfr* mRNA were detected in all cell lines, the *erbb-2* mRNA was induced 23-fold in MDA-MB-468 under hypoxia and dropped down to 1.7-fold under hypoxia compared with the standard conditions (Figure 3). In MDA-MB-231, the *erbb-2* mRNA was reduced to 0.09-fold compared with standard culture conditions and recovered to a value of 1.94 upon reoxygenation. A completely different response was observed in BC-M1, and the *erbb-2* mRNA reached a value of 1.85 under hypoxia and dropped down to a value of 0.18-fold compared with the standard culture conditions.

### 3.5. Discordance of mRNA and Protein Levels under Standard Conditions, Hypoxia, and Reoxygenation

The mRNA levels were compared with the corresponding protein levels under hypoxia or reoxygenation. A total of 38 measurements were statistically analyzable, of which 21 reached a *p*-value of 0.05 and lower. Thus, 55% of the analyzed mRNA levels differed significantly from their protein levels (Table 1).

For some gene products, the mRNAs were detected, but not the corresponding protein. The cell lines with mesenchymal attributes—MDA-MB-231 and BC-M1—were positive for the *epcam* mRNA, but we could detect the corresponding protein. Moreover, we found statistically significant lower values for the EpCAM protein compared with the *epcam* mRNA in MDA-MB-468 under hypoxia and reoxygenation and significantly higher protein levels under hypoxia in MCF-7 (Figure 3, Table 1). This finding was even more pronounced for *er-α*, where we could not detect the ER-α protein in BC-M1 but found a statistically significant (*p* = 0.003) downregulation of the *er-α* mRNA under reoxygenation compared with hypoxia (Figure 3, Table 1). In the case of MDA-MB-231, we observed an upregulation of the *er-α* mRNA from 1.7-fold (hypoxia) to 4.0-fold (reoxygenation; *p* = 0.052), but the ER-α protein could not be detected. In MCF-7, where we detected the *er-α* mRNA and the ER-α protein, we found statistically significant discordances between the mRNA and protein levels under hypoxia and reoxygenation (*p* = 0.003 for both cases).

Similar observations were made for PD-L1. We could detect the *pd-l1* mRNA in MCF-7 and MDA-MB-468 but not the corresponding protein. An interesting response of the *pd-l1* gene product was observed in MDA-MB-231 under hypoxia, where the *pd-l1* mRNA was downregulated to 0.56-fold, and the corresponding protein levels were elevated 1.98-fold compared to standard culture conditions, leading to a significant (*p* = 0.029) deviation of the mRNA and protein levels (Figure 3, Table 1). The finding on MDA-MB-231 could be confirmed for BC-M1, where the levels of the *pd-l1* mRNA remained unchanged under hypoxia. However, the induction of the PD-L1 protein increased 3.6-fold compared with standard culture conditions, leading to a significant deviation of the mRNA and protein levels under hypoxia (*p* = 0.022).

In MDA-MB-468, a typical response to hypoxia was observed, where an increase of the *grp78* mRNA levels was observed under hypoxia, while the corresponding protein levels were reduced to 0.49-fold compared with standard conditions; thus the *grp78* mRNA levels differed significantly from the protein levels (*p* = 0.001). Accordingly, under reoxygenation, the *grp78* mRNA was reduced with an increase in the protein level (Figure 3, Table 1). Compared with standard conditions, no changes in the *grp78* mRNA levels were observed under hypoxia or reoxygenation in MCF-7. The Grp78 protein levels under hypoxia and reoxygenation were reduced in MCF-7 to approx. one-third of the standard culture condition. Such a cell line-specific response was observed even within the group of the cells with mesenchymal attributes—MDA-MB-231 and BC-M1. In MDA-MB-231, the *grp78* mRNA levels were downregulated under hypoxia and recovered under reoxygenation, whereas for BC-M1, the opposite response was observed (Figure 3, Table 1).

For the gene products of the ErbB family—EGFR and ErbB-2—significant discordances between mRNA and protein levels were observed. We detected only minor changes for the *egfr* mRNA in response to hypoxia and reoxygenation in the cell lines. In contrast, a strong EGFR induction on protein level was observed in the ER-α positive cell line MCF-7, both under hypoxia and reoxygenation, compared with standard conditions. Similarly, in the basal-like breast cancer cell line—MDA-MB-468—and in the DTC cell line from the bone marrow—BC-M1—the induction of the EGFR protein was observed (Figure 3, Table 1). Even though a member of the same family, the response of the *erbb-2* gene products was completely different from that of *egfr*. For MCF-7, no changes in the *erbb-2* levels were detected on mRNA and protein level. In MDA-MB-468, under hypoxia, induction of the *erbb-2* mRNA to 23-fold of the standard conditions was detected, while the ErbB-2 protein was induced only to 6-fold. This was followed by a decline of the *erbb-2* mRNA to 1.7-fold under reoxygenation, whereas the ErbB-2 protein remained constant under those conditions. A remarkable strong reaction of the *erbb-2* mRNA to hypoxia and reoxygenation was detected in MDA-MB-231 and BC-M1. Here, the *erbb-2* mRNA decreased to 0.09-fold in MDA-MB-231 under hypoxia and to 0.18-fold in BC-M1 under reoxygenation, which were the strongest changes we detected for all gene products (Figure 3, Table 1).

### 3.6. Hypoxia and Reoxygenation Affected the Protein Phosphorylation Status

Changes in the protein phosphorylation can occur very rapidly under altering microenvironmental conditions. During the analyses of the hypoxia response pathways (Section 3.1), we investigated the phosphorylation of p70 S6 kinase on T389, which lied downstream of mTOR. We additionally investigated the phosphorylation of the protein kinase B (AKT), which integrated signals from the EGFR/ErbB-2 axis and the mTOR pathway (Figure 4). The phosphorylation of AKT was strongly induced in MCF-7, both under hypoxia and reoxygenation, whereas in BC-M1, no substantial changes were observed. In contrast, both in MDA-MB-468 and MDA-MB-231, a strong reduction of the phosphorylation of AKT upon reoxygenation was detected. On the other hand, we observed only minor changes in the phosphorylation of the map kinase Erk1/2.

Since the STAT pathway has been implicated in breast cancer metastasis to the bone, we analyzed the activation of STAT3. Here, we found an activation of STAT3 in MCF-7 under hypoxia, whereas in MDA-MB-468, no changes in the activation of STAT3 were detected (Figure 4). On the other hand, both for MDA-MB-231 and BC-M1, strong activation of STAT3 was detected under hypoxia, followed by a decline of the signals upon reoxygenation.

## 4. Discussion

The detection and molecular analysis of CTC in blood have become increasingly important for understanding metastasis biology and the effects of therapies in breast cancer patients [1]. Even though it is well-known that CTC displays an intra-patient heterogeneity [36], it is not clear if the levels of specific gene products detected in CTC accurately reflect the situation in the original primary or metastatic tumor tissue.

The rapid change of the microenvironmental conditions of tumor cells that are released from the tissue into the blood is frequently disregarded and under-investigated. Among other conditions, hypoxia is one crucial stress factor that can massively affect the expression profile of cells at the mRNA and protein levels. If tumor cells disseminate from hypoxic areas of the tumor tissue into the well-oxygenated bloodstream, these cells may experience a reoxygenation pulse. However, it is practically impossible to determine the exact oxygen concentration and time span that tumor cells are subjected to at their original tissue site before their release into the blood. Moreover, the transit time of CTC in the blood circulation cannot be determined for every single cell.

Since CTC with a hypoxic phenotype is detected in breast cancer patients [19], it is reasonable to assume that at least a subset of CTC is subjected to hypoxia prior to dissemination. The disordered architecture in a tumor may result in insufficient oxygen supply, even adjacent to the blood vessels [9]. Moreover, hypoxia-induced cell migration [37] can promote the extravasation of tumor cells from inner sections of a tumor into the blood.

In the present study, we, therefore, developed an in vitro model that represented only an approximation to the in vivo conditions in cancer patients. Selection of 1% O_2_ for 14 days in our study might be regarded as quite severe conditions that only a subpopulation of tumor cells had to cope with prior to dissemination. On the other hand, these conditions might be regarded as an upper limit; effects that were not observed under these conditions would be presumably very rare in vivo. It has to be noted that the abrogation of the global protein synthesis with an accumulation of the mRNAs is an acute hypoxic stress reaction that normally occurs in a time span of hours [17], whereas we focused on the analysis of longer-term effects of hypoxia prior to re-oxygenation in order to model the long-term settlement of tumor cells in tumor tissues before their release into the blood.

Epithelial marker proteins from the keratin family (e.g., CK5, CK8, CK18, CK19) are frequently used for CTC detection using an anti-keratin antibody cocktail [38]. Alternatively, the protein EpCAM is frequently applied for automatized capture and detection of CTC [34]. Here, we found that these epithelial marker proteins remained quite stable under hypoxia and reoxygenation, supporting the view that these markers are suitable for the detection of epithelial CTCs in blood. We detected in tumor cells, with mesenchymal attributes, *epcam* mRNA, but not the corresponding EpCAM protein. However, the epcam mRNA levels were rather low in these cells, suggesting that the corresponding protein levels were below the detection limit of our protein assay.

Beyond marker proteins, we analyzed proteins that are targeted in cancer therapies. EGFR is a therapeutic target in basal-like breast cancer, ErbB-2 is a target in the ErbB-2 overexpressing breast cancers, and ER-α is a target to hormone therapy in luminal breast cancers. In our experiments, hypoxia changed the phenotype of all investigated cell lines to an EGFR/ErbB-2 double-positive phenotype, which characterizes aggressive breast cancers [39]. ErbB-2-positive CTC in patients with ErbB-2 negative tumors has been detected in previous studies and could assist the stratification and monitoring of ErbB-2-directed therapies [40].

Immune checkpoint therapies have led to fundamental changes in cancer therapy, and CTCs have become subject of recent investigations [41]. In this context, PD-L1 is currently one of the most prominent therapeutic targets, which is also expressed on CTCs [41]. Here, we consistently detected the highest protein levels for the PD-L1 protein in EpCAM-negative/keratin-weakly positive tumor cells with mesenchymal attributes. Keratins or EpCAM have been, however, frequently used as detection markers for CTCs in studies on PD-L1 assessment [34,42]. Our experiments were consistent with previous reports, demonstrating that PD-L1 positive CTC undetectable by EpCAM or keratins exist in cancer patients [41]. The bone marrow-derived DTC cell line BC-M1, used as a model, exemplified that breast cancer cells with mesenchymal attributes were able to reach secondary sites, display high PD-L1 levels in the ground state, and even increase the PD-L1 levels under hypoxia. In previous studies, we could demonstrate a marked intra-patient and inter-patient heterogeneity of PD-L1 expression in CTCs from patients with breast cancer [34] and other solid tumors [43,44]. Furthermore, BC-M1 strongly induced EGFR under hypoxia, resulting in an EGFR/PD-L1 double-positive phenotype. In head and neck cancer patients, a CTC cluster with such a phenotype has been detected [45]. However, our present study indicated that PD-L1-positive tumor cells present in hypoxic tissues might downregulate PD-L1 after reoxygenation in the blood, suggesting a novel mechanism of resistance for CTCs against PD-L1 therapy.

It was previously reported that cycling hypoxia/reoxygenation selects for cells with a quiescent state with downregulated HIF-1α and p70 S6 kinase [46]. While we confirmed the downregulation of HIF-1α under hypoxia in the present study, we also observed a strong induction of STAT3 phosphorylation and reduction and p70 S6 kinase phosphorylation after 14 days of hypoxia, which might point towards the acquisition of a quiescent state. However, we could easily propagate all investigated cell lines throughout the 14 days of hypoxia, and previous experiments have shown that the cells continue to proliferate under hypoxia [14]. Indeed, we observed a strong induction of HIF-1α and reduced phosphorylation of STAT3 after reoxygenation, suggesting that the cells could easily pass over to an active state when the microenvironmental conditions change. Thus, one can speculate that the release of tumor cells into the blood may activate cells and increase their ability to survive and extravasate.

The mRNA and protein levels were uncoupled under hypoxia for some gene products, and the uncoupling was maintained under reoxygenation, affecting Grp78, EGFR, ErbB-2, and ER-α. Many mechanisms may account for discrepancies between mRNA and protein levels [15,47]. Among those, the cap-independent protein translation allows the specific translation of mRNAs that are essential for the survival of tumor cells under hypoxia when the global protein translation is shut down. For EGFR, such a mechanism has been recently reported [48]. This explained well the high EGFR protein levels observed under hypoxia as compared to the moderate mRNA levels, in particular, in the DTC cell line BC-M1, suggesting that this mechanism might be of potential relevance in tumor cell dissemination. Under normal conditions, the median half-life of proteins is 46 h, and the half-life of mRNA is 9 h [49]. We did not investigate the impact of reoxygenation on the half-lives of the transcripts and proteins. Thus, we could not exclude that some of the effects that we detected might be rather due to accelerated degradation than regulation effects.

## 5. Conclusions

Our data suggested that CTCs in the blood might show specific changes in certain mRNA or protein levels (e.g., PD-L1), which needs to be taken into account when CTC analysis is used in liquid biopsy to gain insights into primary tumors or metastases. The present data further suggested that reoxygenation might activate tumor cells released into the blood, but reoxygenation might also lower PD-L1 levels. Thus, CTC might be more vulnerable to T-cell-mediated lysis, which supports the concept that the dissemination of tumor cells through the bloodstream might be the “Achilles heel” of cancer metastasis.

## Figures and Tables

**Figure 1 cells-09-01316-f001:**
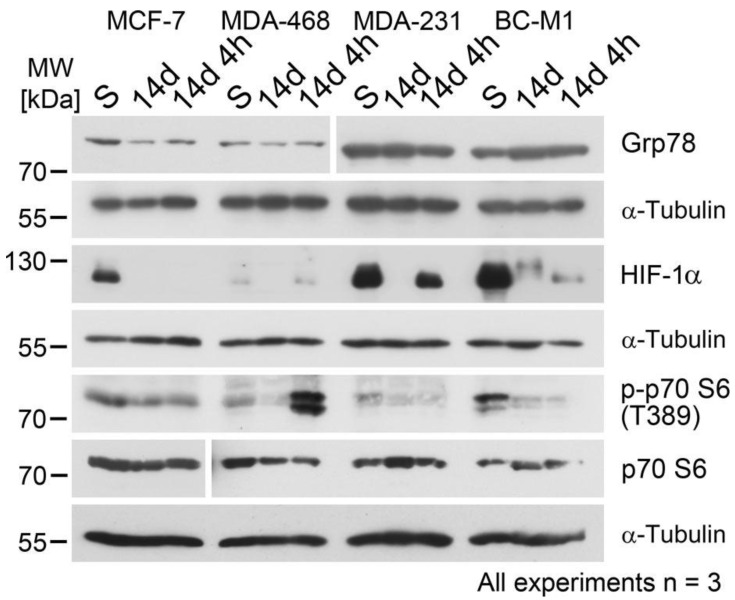
Investigation of the cellular response to hypoxia and reoxygenation to the three major hypoxia response programs by Western blot. Grp78 (78 kDa glucose-regulated protein) of the unfolded protein response, HIF-1α (hypoxia-inducible factor 1-alpha) as the master regulator of metabolic adaptation to hypoxia, and the phosphorylation of the p70 S6 kinase of the mTOR (mammalian target of rapamycin) pathway were analyzed. The cells were cultured under standard conditions for 14 days (S), under hypoxia (1% O_2_) for 14 days (14 d), and after cultivation under hypoxia, followed by reoxygenation (10% O_2_) for 4 h (14 d 4 h). Three biological replicates were analyzed for each cell line. Grp78 was analyzed together with the proteins shown in Figure 2; therefore, the image for alpha-tubulin is the same as in Figure 2. S: standard cell culture conditions, 14d: hypoxia, 14d 4h: reoxygenation.

**Figure 2 cells-09-01316-f002:**
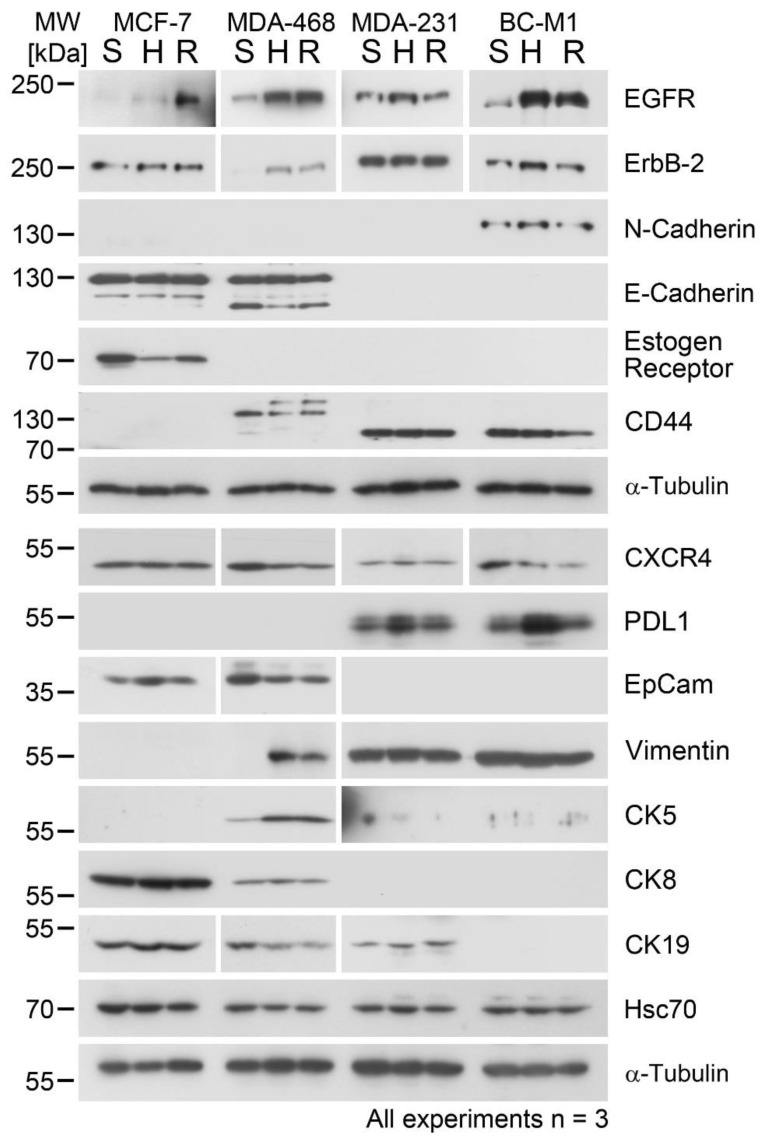
Western blot analysis for the denoted proteins and cell lines after cultivation under standard cell culture conditions for 14 days (S), under hypoxia (1% O_2_) for 14 days (H), and after cultivation under hypoxia followed by reoxygenation (10% O_2_) for 4 h (R). Three biological replicates were analyzed for each cell line. kDa: kilodaltons, S: standard cell culture conditions, H: hypoxia, R: reoxygenation.

**Figure 3 cells-09-01316-f003:**
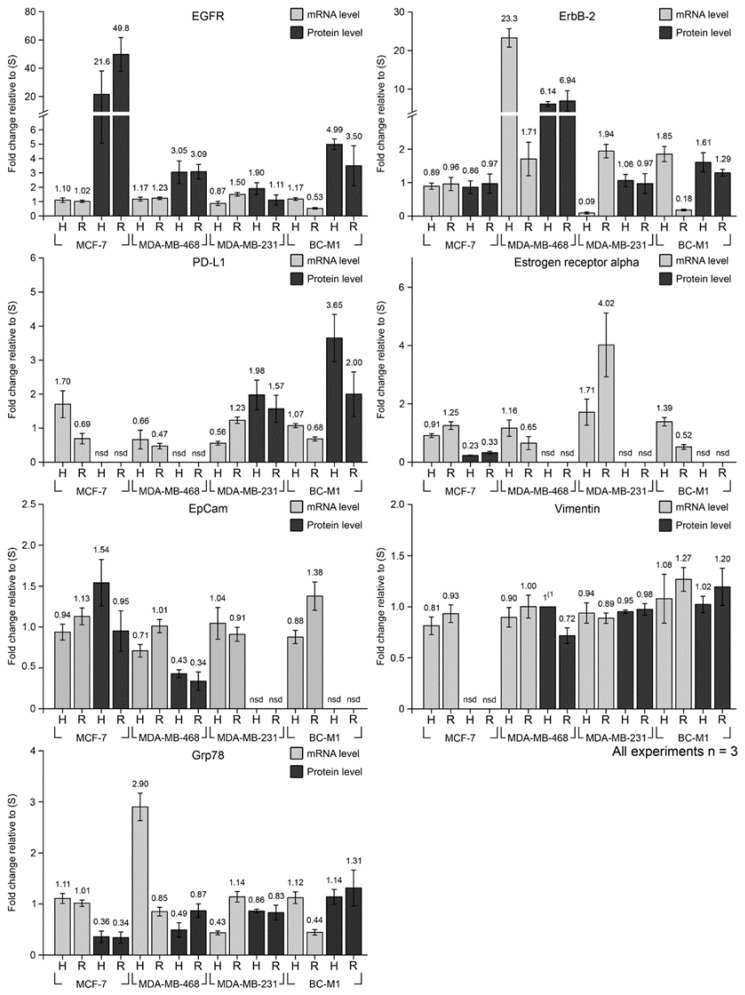
Detected mRNA and protein levels, depending on the oxygenation status. Analyzed were the cultivation under hypoxia (1% O_2_) for 14 days (H) and cultivation under hypoxia followed by reoxygenation (10% O_2_) for 4 h (R). Standard cell culture conditions for 14 days were reference values and set as 1.0. The error bars show the standard deviation from the mean value (given as numbers) of three biological experiments; nsd: no signal detected; (1: For vimentin in MDA-MB-468, the H value was set as 1.0 since under standard culture conditions no signal was detectable).

**Figure 4 cells-09-01316-f004:**
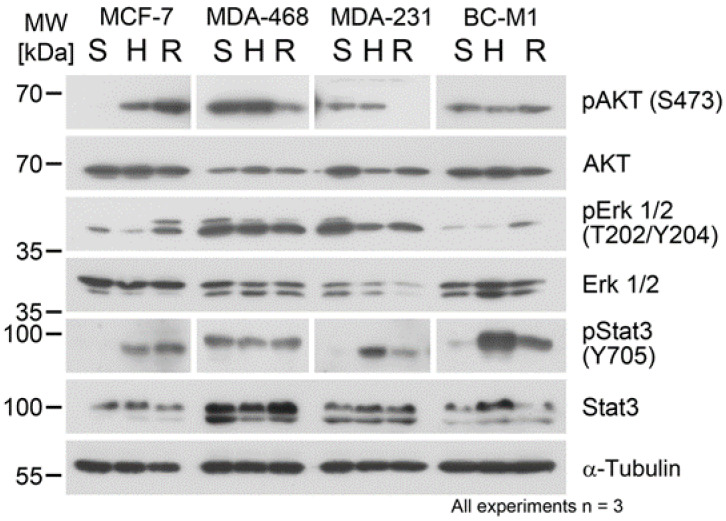
Western blot analysis for the denoted proteins and their phosphorylation status after cultivation of the cell lines under standard cell culture conditions for 14 days (S), under hypoxia (1% O_2_) for 14 days (H), and after cultivation under hypoxia for 14 days followed by reoxygenation (10% O_2_) for 4 h (R). Three biological replicates were analyzed for each cell line.

**Table 1 cells-09-01316-t001:** Details for the denoted gene products. Values were adjusted to the value of the standard cell culture condition, which was set to 1.0. Hypoxia for 14 days (Hyp), hypoxia for 14 days followed by reoxygenation (Reox). Three biological samples were analyzed for each cell line. NA: not analyzable. Vs.: versus.

Gene Product	Cell Line	ΔCq Under S (mRNA)	Condition	x-Fold Change (mRNA)	STDEV (mRNA)	x-Fold Change (Protein)	STDEV (Protein)	*p*-Value Protein Level vs. mRNA Level	*p*-Value Hyp vs. Reox (Protein Level)	*p*-Value Hyp vs. Reox (mRNA Level)
PD-L1	MDA-MB-231	6.73	Hyp	0.56	0.06	1.98	0.44	0.029	0.004	0.045
			Reox	1.23	0.09	1.57	0.40	0.277		
	MDA-MB-468	13.97	Hyp	0.66	0.27	no signals		NA	NA	0.347
			Reox	0.47	0.08	no signals		NA		
	MCF-7	16.52	Hyp	1.70	0.40	no signals		NA	NA	0.036
			Reox	0.69	0.15	no signals		NA		
	BC-M1	6.52	Hyp	1.07	0.06	3.65	0.69	0.022	0.024	0.001
			Reox	0.68	0.06	2.00	0.66	0.073		
Vimentin	MDA-MB-231	−2.78	Hyp	0.94	0.10	0.95	0.02	0.880	0.422	0.496
			Reox	0.89	0.05	0.98	0.06	0.119		
	MDA-MB-468	7.11	Hyp	0.90	0.10	1.00 ^1^	0	NA	0.023	0.309
			Reox	1.00	0.11	0.72	0.08	NA		
	MCF-7	8.26	Hyp	0.81	0.09	no signals		NA	NA	0.178
			Reox	0.93	0.09	no signals		NA		
	BC-M1	−3.30	Hyp	1.08	0.24	1.02	0.08	0.715	0.237	0.309
			Reox	1.27	0.12	1.20	0.18	0.609		
EGFR	MDA-MB-231	2.69	Hyp	0.87	0.15	1.90	0.41	0.037	0.028	0.006
			Reox	1.50	0.13	1.11	0.35	0.185		
	MDA-MB-468	0.61	Hyp	1.17	0.14	3.05	0.79	0.050	0.887	0.572
			Reox	1.23	0.09	3.09	0.50	0.021		
	MCF-7	8.40	Hyp	1.10	0.16	21.58	16.50	0.165	0.227	0.496
			Reox	1.02	0.08	49.76	11.89	0.019		
	BC-M1	3.51	Hyp	1.17	0.10	4.99	0.38	0.002	0.133	0.002
			Reox	0.53	0.05	3.50	1.39	0.066		
EpCAM	MDA-MB-231	4.33	Hyp	1.04	0.19	no signals		NA	NA	0.366
			Reox	0.91	0.09	no signals		NA		
	MDA-MB-468	−0.21	Hyp	0.71	0.08	0.43	0.05	0.011	0.178	0.010
			Reox	1.01	0.08	0.34	0.11	0.002		
	MCF-7	0.80	Hyp	0.94	0.10	1.54	0.28	0.052	0.089	0.081
			Reox	1.13	0.10	0.95	0.25	0.341		
	BC-M1	10.65	Hyp	0.88	0.08	no signals		NA	NA	0.022
			Reox	1.38	0.17	no signals		NA		
ErbB2	MDA-MB-231	8.91	Hyp	0.09	0.03	1.06	0.18	0.010	0.370	0.003
			Reox	1.94	0.20	0.97	0.30	0.013		
	MDA-MB-468	13.42	Hyp	23.26	2.39	6.14	0.64	0.004	0.618	0.003
			Reox	1.71	0.50	6.94	2.66	0.072		
	MCF-7	7.19	Hyp	0.89	0.09	0.86	0.19	0.822	0.199	0.622
			Reox	0.96	0.20	0.97	0.29	0.963		
	BC-M1	6.44	Hyp	1.85	0.23	1.61	0.29	0.327	0.247	0.006
			Reox	0.18	0.03	1.29	0.10	0.001		
ERα	MDA-MB-231	19.09	Hyp	1.71	0.45	no signals		NA	NA	0.052
			Reox	4.02	1.10	no signals		NA		
	MDA-MB-468	14.55	Hyp	1.16	0.28	no signals		NA	NA	0.072
			Reox	0.65	0.22	no signals		NA		
	MCF-7	4.59	Hyp	0.91	0.07	0.23	0.01	0.003	0.072	0.027
			Reox	1.25	0.13	0.33	0.05	0.003		
	BC-M1	13.12	Hyp	1.39	0.15	no signals		NA	NA	0.003
			Reox	0.52	0.08	no signals		NA		
Grp78	MDA-MB-231	1.83	Hyp	0.43	0.04	0.86	0.04	<0.001	0.672	0.003
			Reox	1.14	0.10	0.83	0.15	0.003		
	MDA-MB-468	5.92	Hyp	2.90	0.27	0.49	0.14	0.001	0.085	0.003
			Reox	0.85	0.08	0.87	0.13	0.834		
	MCF-7	3.63	Hyp	1.11	0.10	0.36	0.11	0.001	0.903	0.227
			Reox	1.01	0.06	0.34	0.11	0.002		
	BC-M1	2.40	Hyp	1.12	0.11	1.14	0.15	0.862	0.295	0.003
			Reox	0.44	0.05	1.31	0.35	0.047		

^1^ no signals were detected for the standard cell culture condition. Therefore, this value was set as the reference value of 1.0.

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
