# Peer review of "In Vitro Modeling of Reoxygenation Effects on mRNA and Protein Levels in Hypoxic Tumor Cells upon Entry into the Bloodstream"

_cells, 2020, doi:10.3390/cells9051316_

Round 1
Reviewer 1 Report
The data added in this revision does not connect with the rest of the story. The conclusions are still poorly supported. The manuscript is now extremely long and although it brings some biological observations relative to mRNA/protein expression in chronic hypoxia/reoxygenation, it does not bring a meaningful well-corroborated biological conclusion. It is extremely important to include blots of HIF-1a and HIF-2a when studying hypoxia/reoxygenation effects.
Reviewer 2 Report
The authors have now addressed my comments.
Round 2
Reviewer 1 Report
No comments